# Correlative Multi-Modal Microscopy: A Novel Pipeline for Optimizing Fluorescence Microscopy Resolutions in Biological Applications

**DOI:** 10.3390/cells12030354

**Published:** 2023-01-17

**Authors:** Simone Pelicci, Laura Furia, Pier Giuseppe Pelicci, Mario Faretta

**Affiliations:** 1Department of Experimental Oncology, European Institute of Oncology IRCCS, 20139 Milan, Italy; 2Department of Oncology and Hemato-Oncology, University of Milan, 20122 Milan, Italy

**Keywords:** multi-modal microscopy, fluorescence microscopy, single molecule localization microscopy, confocal microscopy, image cytometry, image analysis

## Abstract

The modern fluorescence microscope is the convergence point of technologies with different performances in terms of statistical sampling, number of simultaneously analyzed signals, and spatial resolution. However, the best results are usually obtained by maximizing only one of these parameters and finding a compromise for the others, a limitation that can become particularly significant when applied to cell biology and that can reduce the spreading of novel optical microscopy tools among research laboratories. Super resolution microscopy and, in particular, molecular localization-based approaches provide a spatial resolution and a molecular localization precision able to explore the scale of macromolecular complexes in situ. However, its use is limited to restricted regions, and consequently few cells, and frequently no more than one or two parameters. Correlative microscopy, obtained by the fusion of different optical technologies, can consequently surpass this barrier by merging results from different spatial scales. We discuss here the use of an acquisition and analysis correlative microscopy pipeline to obtain high statistical sampling, high content, and maximum spatial resolution by combining widefield, confocal, and molecular localization microscopy.

## 1. Introduction

Countless processes occur in cells and tissues in a way that depends on the structures of different cellular compartments. In modern biomedical research, optical imaging has been widely used to obtain structural and functional information at the cellular, subcellular, and molecular levels [1]: different types of optical imaging techniques exist to investigate each event in relation to the properties of the target compartments and molecules. In cell biology, standard fluorescence microscopy provides one of the most powerful tools to study cellular structures and functions in living and fixed samples. Nonetheless, the resolution achieved is still much less than that provided by Electron Microscopy (EM) (~1 nm). However, EM analysis shows limitations compared to fluorescence: it is not compatible with living cell observation, it requires complex sample preparation procedures, and it has low statistical output and a limited field of view. Consequently, the study of rare events in cells or tissues is extremely challenging and time-consuming, if feasible. To fill this gap, several laboratories have combined fluorescent cell imaging and EM, developing one of the most known multi-modal correlative approaches, i.e., Correlative Fluorescence and Electron Microscopy [2,3,4]. It employs the molecular specificity of fluorescence microscopy, providing high contrast imaging of molecular targets. Antibody and/or fluorescent proteins labellingenables their localization for the EM analysis: an unsurpassed resolution can be reached, but still with a limited throughput.

The recent improvement in the resolving power with the advent of super resolution imaging (~20 nm) [5,6,7] and the technical challenges due to the sample preparation for EM (e.g., sample dehydration, fixation and slicing) has made the combination of optical fluorescence techniques very attractive (e.g., correlative confocal and super resolution microscopy) [8], also for the possibility of simplifying the experimental procedures, thanks to the homogenous and easy requirements in sample processing.

Moreover, in optical fluorescence imaging, the progress of high-content microscopy (HCM), combined with advanced computational image analysis, has enabled a collection of multi-parametric datasets with high quantitative information on biological systems from low to high optical resolution [9,10]. Automated microscopy, together with high-content approaches, has extended the number of simultaneously analyzable parameters at the single-cell level, identifying a plethora of quantitative phenotypic profiles [11,12,13,14]. In recent years, innovative techniques have been implemented to increase the information content in fixed or living cells. One of these strategies is represented by the multiplexing approach, which allows the visualization of multiple targets by sequential immunostainings, thus taking over the limits of classical detection by fluorochromes with non-overlapping spectral profiles [15,16]. Multiplex immunostaining has become a powerful tool for single-cell and tissue-section identification of neoplastic, reactive, inflammatory, and normal phenotypes [17,18].

Despite the ability to simultaneously extract hundreds of biologically informative measurements from single-cell imaging, sometimes, fluorescence microscopy, in particular high resolution fluorescence microscopy, lacks a consistent statistical sampling, limiting the analysis to a small number of cells. In quantitative high-content and -resolution imaging, the extension of single-cell multiparameter analysis to hundreds and thousands of events has always been a challenging task [19,20,21]. Moreover, inside a cell population, the simultaneous identification, quantification, and visualization of different phenotypes in a statistically relevant way would be crucial to increase the understanding of biological heterogeneity [22,23]. Microscope automation, together with image cytometry, can combine high-content imaging measurements with a statistically relevant number of analyzed events, producing high-resolution imaging of the inner cell compartments, by conventional and confocal automated microscopes [24].

Despite the versatility of the ensemble of the modern fluorescence microscopies, optical imaging obeys a sort of indetermination principle: the optimization of a selected performance, i.e., spatial resolution, temporal resolution, sensitivity, or statistical sampling inevitably impacts on the others. For example, to reach an increased spatial resolution, such as the one reached by super resolution microscopy, image collection must satisfy the Nyquist theorem, thus reducing the physical pixel dimensions. Since the higher the number of pixels requested to image a certain region, the lower the number of collected photons from a smaller pixel, both temporal resolution and signal-to-noise ratio are negatively affected. The high localization precision and single-molecule sensitivity of Single-Molecule Localization Microscopy (SMLM) require the acquisition of thousands of frames [25], which has made this technology hardly suitable for live cell imaging or incompatible with high statistical sampling.

Addressing a specific question during the execution of an imaging experiment should thus lead to the switching from one technique to the other. In its classic definition, a correlative microscopy experiment was almost exclusively intended as the combination of different technologies aimed at the maximal spatial resolution, e.g., the combination of electron or atomic force microscopy and fluorescence microscopy [26,27,28].

An extended definition of correlative microscopy can be formulated: linking data from multi-modal acquisitions on the same sample, each one of them aided at the maximization of a certain resolution. Luckily, all the fluorescence microscopy derivatives can be frequently added as modules to the single core represented by the classical widefield fluorescence microscope, making possible the desired correlation among the different data. However, few examples are present in the literature that have reported how to combine optical microscopy techniques on the same instrument to perform variable-resolution imaging. Examples of correlative optical microscopies were focused on the combination of different modalities provided by nonlinear microscopy (two-photon excitation, second harmonic generation, spectral imaging) [29,30,31,32].

Automation is absolutely instrumental to combine and multiplex different microscopy approaches, e.g., to monitor events in living cell microscopy and observe the targets in high-content, super-resolution microscopy [33]. Automated microscopy for image cytometry provides another example of how this goal can be achieved [24,34,35]. High-resolution widefield microscopy can supply quantitative and “temporal” information, both at cellular and intracellular resolutions, to study selected phenotypes with high statistical sampling and sensitivity, e.g., the expression of molecular markers to activate cell-cycle checkpoints in response to external insults [36]. Simultaneously, as frequently observed in correlative studies between fluorescence and EM, the measured data can redirect a second 3D high-resolution analysis on targeted single cells by confocal microscopy revealing novel regulatory mechanisms based on the spatial and temporal re-localization of the involved proteins.

The same approach can be extended to new experimental pipelines for histological multiplexed spatial-proteomics analysis associated with progressively rescaled resolution. A set of specific computational tools can be employed to identify areas of interest in tissue-samples and tumor micro-array (TMA) by large statistical-sampling widefield imaging at increasing resolution and sensitivity. The resolving power of confocal imaging is then applied to obtain a 3D reconstruction of the targeted regions [37].

Here, we extended the concept of Correlative Multi-Modal Microscopy, describing a workflow that associates fast widefield imaging of thousands of cells to a combination of high- and super-resolution microscopy techniques. First, a region enclosing a statistically relevant number of objects is acquired by an automated widefield microscope. Multi-parameter quantitative imaging of thousands of cells is performed in a reasonable time, by using an automated-acquisition protocol, allowing the selection and physical retrieval of cell subpopulations. Subsequently, the physical coordinates of every targeted cell are automatically recalculated for a deeper investigation in high- and super-resolution analysis, e.g., linking the cell position in the cell-cycle phase with the single-molecule resolution of molecular interactions. Examples of spatially correlated multi-modal optical techniques are shown, including (i) 2D-dSTORM and 3D-confocal imaging, and (ii) TIRF and 3D-confocal imaging. Single-molecule information of dSTORM combined with confocal imaging improved the spatial resolution and extended the number of simultaneously analyzed parameters, detecting until four target proteins at variable resolution. Simultaneous TIRF and confocal imaging instead provide a 3D view of the cell space associated with a focused high-resolution analysis of the glass-cell interface plane (basal membrane). As proof of concept, we applied the Correlative Multi-Modal Microscopy pipeline to characterize localization-dependent molecular complexes of the 53BP1 protein: analysis of subcellular localization and identification of site-specific molecular partners can be performed thanks to the employment of correlative high-resolution multiparameter widefield, 3D confocal, Structured Illumination, total internal reflection, and STORM microscopy.

## 2. Materials and Methods

### 2.1. Cell Culture

MCF10A cells were cultured in 50% DMEM High Glucose with stable L-glutamin (DMEM) (Euroclone, Milan, Italy) + 50% Ham’s F12 Medium (ThermoFisher Scientific, Waltham, MA, USA) containing 5% Horse serum, 50 ng/mL Penicillin/Streptomycin (both from Euroclone), 50 ng/mL Cholera Toxin (Merck Life Science, Milan, Italy), 10 µg/mL Insulin (Merck Life Science, Milan, Italy), 500 ng/mL Hydrocortisone (Merck Life Science, Milan, Italy), and 20 ng/mL EGF (Pepro Tech, Cranbury, NJ, USA) at 37 °C in 5% CO_2_. Cells were grown on glass bottom culture dishes (MatTek, Ashland, MA, USA). When the cells reached 70% confluence, they were fixed for 10 min in 4% paraformaldehyde (wt/vol) to guarantee exponential growth. In some experiments, Ethinyl-deoxyUridine (EdU) (ThermoFisher Scientific, Waltham, MA, USA) was added to the culture media at 10 µM final concentration. Cells were incubated for 20 min and then fixed as specified above.

### 2.2. Immunofluorescence of MCF10A Cells

Fixed MCF10A cells were washed and permeabilized for 10 min in a permeabilization buffer containing 0.1% Triton X-100 (*v*/*v*) in PBS. Samples were processed according to different types of experiments.

#### 2.2.1. High Content Widefield/3D-Confocal Microscopy

After PLA staining (see Section 2.4), samples were first briefly fixed in 4% paraformaldehyde (*wt*/*v*) to crosslink the antigen–antibodies complexes, limiting their dissociation, and incubated for 1 h with the following antibodies diluted in 5% BSA in PBS: mouse monoclonal IgG3 anti Lamin A (MA1-06101, ThermoFisher Scientific, Waltham, MA, USA) and rat monoclonal anti-Tubulin (MAB1864, Chemicon, Merck Life Science, Milan, Italy). Cells were washed 3 times in PBS and incubated for 45 min at room temperature with the following secondary antibodies: Alexa Fluor^®^ 790 AffiniPure Donkey Anti-Mouse IgG3 (H + L) (115-655-209, Jackson-immunoresearch, West Grove, PA, USA), Alexa Fluor^®^ 594 AffiniPure Donkey Anti-Rat IgG (H + L) (712-585-153, Jackson-immunoresearch, West Grove, PA, USA). Phalloidin Atto 425 (66939, Merck Life Science, Milan, Italy) was added to the solution to mark Actin filaments. Cells were then briefly crosslinked in 4% paraformaldehyde (*wt*/*v*) to stabilize antibody complexes and incubated overnight with the following directly conjugated antibodies (5% BSA in PBS): Alexa488-conjugated mouse anti phosphoH2A.X (ser39) (γH2A.X) (613406, Biolegend, San Diego, CA, USA), and anti-human KI67 Horizon V450-conjugated (561281, BD Biosciences, Franklin Lakes, NJ, USA). After washing and incubation with Hoechst 33,342 (H3570, ThermoFisher Scientific, Waltham, MA, USA), samples were mounted in SlowFade™ Gold Antifade Mountant for fluorescence microscopy analysis (ThermoFisher Scientific, Waltham, MA, USA).

#### 2.2.2. High Content Widefield/3D-SIM Microscopy

After EdU staining (see Section 2.3), samples were processed for PLA detection. After washing and incubation with Hoechst 33342 (H3570, ThermoFisher Scientific, Waltham, MA, USA), samples were mounted in SlowFade™ Gold Antifade Mountant for fluorescence microscopy analysis (ThermoFisher Scientific, Waltham, MA, USA).

#### 2.2.3. Correlative 3D-Confocal/STORM Microscopy

Samples were incubated for 1 h with the following antibodies diluted in 5% BSA in PBS: rabbit polyclonal anti-53BP1 (ab36823, Abcam, Cambridge, UK), and mouse monoclonal IgG2b DynLL1/Pin1 (MAB2294, R&D systems, Minneapolis, MN, USA). Cells were washed 3 times in PBS and incubated for 45 min at room temperature with the following secondary antibodies: Alexa Fluor^®^ 647 AffiniPure Donkey Anti-Rabbit IgG (H + L) (711-605-152, Jackson-immunoresearch, West Grove, PA, USA) and Cy™3 AffiniPure Donkey Anti-Mouse IgG (H + L) (715-165-150, Jackson-immunoresearch, West Grove, PA, USA). Cells were then briefly re-fixed in 4% paraformaldehyde (wt/vol) and incubated overnight with the following primary antibodies: Alexa488-conjugated mouse anti phosphoH2A.X (ser39) (γH2A.X) (613406, Biolegend, San Diego, CA, USA) and rat monoclonal anti-Tubulin (MAB1864, Chemicon, Merck Life Science, Milan, Italy). Cells were then rinsed 3 times in PBS and incubated for 1 h with donkey anti-Rat IgG (H + L) biotin (a18749, ThermoFisher Scientific, Waltham, MA, USA) and for 45 min at room temperature with streptavidin Atto425 conjugated (S000-51, Rockland Immunochemicals, Pottstown, PA, USA). After washings, DNA was counterstained with Chromomycin, and finally, cells were briefly re-fixed in 4% paraformaldehyde (*wt*/*v*). Before image acquisition, PBS was replaced by STORM Imaging Buffer.

#### 2.2.4. High Content Widefield (53BP2)/STORM Microscopy

Samples were incubated for 30 min in a blocking solution, 5% BSA (*wt*/*v*) in PBS, and for 1 h with the following antibodies diluted in 5% BSA in PBS: mouse monoclonal IgG1 53BP2 (sc-53861, Santa-Cruz Biotechnologies, Dallas, TX, USA), and rabbit polyclonal 53BP1 (ab36823, Abcam, Cambridge, UK). Cells were washed 3 times in PBS and incubated for 45 min at room temperature with the following secondary antibodies: Alexa Fluor^®^ 647 AffiniPure Goat Anti-Mouse IgG1 (H + L) (115-605-205, Jackson-immunoresearch, West Grove, PA, USA), Cy™3 AffiniPure Donkey Anti-Rabbit IgG (H + L) (711-165-152, Jackson-immunoresearch, West Grove, PA, USA). After washings, samples were briefly re-fixed in 4% paraformaldehyde (*wt*/*v*) and then incubated for 1 h with rat monoclonal anti-Tubulin (MAB1864, Chemicon, Merck Life Science, Milan, Italy), followed by washes and incubation with Alexa Fluor^®^ 488 AffiniPure Donkey Anti-Rat IgG (H + L). After washing and incubation with Hoechst 33342 (H3570, ThermoFisher Scientific, Waltham, MA, USA), cells were finally re-fixed in 4% paraformaldehyde (*wt*/*v*) and put in PBS. Before image acquisition, PBS was replaced by STORM Imaging Buffer.

#### 2.2.5. Correlative 3D-Confocal/TIRF Microscopy

Samples were incubated for 30 min in a blocking solution, 5% BSA (wt/vol) in PBS, and for 1 h with the following antibodies diluted in 5% BSA in PBS: mouse monoclonal IgG2b DynLL1/Pin1 (MAB2294, R&D systems, Minneapolis, MN, USA), rabbit polyclonal 53BP1 (ab36823, Abcam, Cambridge, UK). Cells were washed 3 times in PBS and incubated for 45 min at room temperature with the following secondary antibodies: Alexa Fluor^®^ 647 AffiniPure Donkey Anti-Mouse IgG2b (H + L) (115-605-207, Jackson-immunoresearch, West Grove, PA, USA), and Cy™3 AffiniPure Donkey Anti-Rabbit IgG (H + L) (711-165-152, Jackson-immunoresearch, West Grove, PA, USA). Cells were rinsed 3 times in PBS, briefly re-fixed in 4% paraformaldehyde (*wt*/*v*), and incubated with rat monoclonal anti-Tubulin (MAB1864, Chemicon, Merck Life Science, Milan, Italy). After washings, samples were incubated with Alexa Fluor^®^ 488 AffiniPure Donkey Anti-Rat IgG (H + L) and Phalloidin-Atto425 (66939, Sigma-Aldrich, Merck Life Science, Milan, Italy). Cells were then washed 3 times in PBS, DNA was counterstained with Hoechst 33342, and finally briefly re-fixed in 4% paraformaldehyde (*wt*/*v*) and put in PBS.

### 2.3. EdU Detection

EdU incorporation into DNA was detected using the Click-iT EdU Pacific Blue assay kit (C10418, ThermoFisher Scientific, Waltham, MA, USA), according to the manufacturer’s instructions. All the steps of the Click-iT reaction were performed at room temperature (RT).

### 2.4. In Situ Proximity Ligation Analysis (PLA)

Samples were processed for in situ PLA using the DuoLink in-situ Orange detection-reagent (Sigma-Aldrich, Merck Life Science, Milan, Italy) according to the manufacturer’s instructions. The primary antibodies employed in the PLA assay were rabbit polyclonal anti-53BP1 (ab36823, Abcam, Cambridge, UK) and mouse monoclonal IgG2b DynLL1/Pin1 (MAB2294, R&D systems, Minneapolis, MN, USA). Then, after execution of the PLA reactions leading to the creation of the DNA rolling circles that mark the interaction, cells were incubated with fluorochrome-conjugated secondary antibodies: Pacific Orange Goat anti-Rabbit IgG (H + L) (P31584, ThermoFisher Scientific, Waltham, MA, USA) for High Content Widefield/3D-Confocal Microscopy, or Alexa Fluor^®^ 488 AffiniPure Donkey Anti-Rabbit IgG (H + L) (711-545-152, Jackson-immunoresearch, West Grove, PA, USA) for High Content Widefield/3D-SIM Microscopy, and, in both cases, Alexa Fluor^®^ 647 AffiniPure Donkey Anti-Mouse IgG2b (H + L) (115-605-207, Jackson-immunoresearch, West Grove, PA, USA). Fluorochrome conjugated secondary antibodies, competing with the residual fraction of oligo-conjugated ones not involved in the PLA circle formation, allowed the detection of the cell expression levels of the targeted molecules in addition to the interaction spots.

### 2.5. Microscope Setup

All data were acquired with a commercial Nikon Eclipse Ti2 inverted microscope (Nikon instruments, Tokyo, Japan), equipped with A1R confocal scanhead, N-SIM and N-STORM modules (Nikon instruments, Tokyo, Japan). The fully motorized automated microscope was controlled by the NIS Elements software (version 5.42.01, Laboratory Imaging s.r.o, Praha, Czech Republic). The system performs multicolor widefield, confocal, and single-molecule localization imaging thanks to a pE-4000 (CoolLED, Andover, UK) light source with 16 selectable LED wavelengths (Widefield microscopy) and a LU-NV laser unit (Nikon instruments, Tokyo, Japan) equipped with 5 laser lines (405 nm (23.1 mW), 440 nm (25.5 mW) 488 nm (79.1 mW), 561 nm (79 mW), and 647 nm (137 mW)) (Confocal, SIM, TIRF and dSTORM microscopy). The microscope is equipped with fluorescence filter sets for the optimal detection of the employed fluorochromes thanks to a double-layer turret allowing the combination of 5 cubes per layer. The upper turret is devoted to widefield and STORM imaging, with the lower one acting as an additional filter wheel. SIM images were instead collected through the lower layer by spatially calibrated filter cubes. Emitted light was collected by a CMOS camera (Dual ORCA Flash 4.0 Digital CMOS camera C13440, Hamamatsu, Tokio, Japan) set on a 16-bit scale detection modality (Widefield/SIM/TIRF/dSTORM).

#### 2.5.1. Widefield Microscopy

For widefield imaging, data were acquired using a 60× Plan Apo 1.4 NA objective. Optimal exposure time was set for each fluorescence channel by maximizing the dynamic range and avoiding saturation based on a preliminary observation of randomly chosen fields. In some experiments, a Relay lens (0.4×) was inserted into the optical path to increase the field of view of a 100× 1.49 NA objective gaining high statistical sampling with maximal resolution.

#### 2.5.2. Confocal Microscopy

Confocal data were acquired with Nikon’s A1R confocal microscope with a 60× Plan Apo 1.4 NA and with a 100× 1.49 NA Apochromat objective. Images were collected sequentially using a spectral GaAsP detector unit operating in the 400–720 nm interval and set for optimal detection conditions that maximize the signal-to-noise ratio and limit crosstalk. In the 9-channel imaging experiments, closely emitting fluorochromes (i.e., Atto 425-Phalloidin, and Horizon v450KI67; Star Orange-PLA and Alexa 594-Tubulin) were detected simultaneously being localized in spatially separated cell compartments. Linear spectral un-mixing was then applied to obtain single-channel imaging. High-speed scanning was performed thanks to the resonance mirrors with a frame rate ranging from 4 to 15 frames per second.

In Correlative 3D-Confocal/2D-dSTORM acquisitions were sequentially performed for each fluorophore with the high-speed galvanometric mirrors with 4X Line Average to maximize image quality in a reduced time. Pinhole size has been set to 0.8 Airy Unit for every collected channel. The image size was set to 512 × 512 pixels, with a pixel size of ~0.07 μm.

In Correlative 3D-Confocal/TIRF acquisitions were sequentially performed for each fluorophore in resonant scanning mode with 2X Line Average to improve image quality. Pinhole size was set to 0.8 Airy Unit for every collected channel. The ROI size was set to 1024 × 1024 pixels, with a pixel size of ~0.06 μm.

In both the modalities, the scanning pixel dwell-time and laser power were set to limit photobleaching effects during Z-stack acquisition. As previously mentioned, the slit aperture of the spectral detection system was set to maintain an optimal Signal-to-Noise ratio avoiding crosstalk among the different channels or performing spectral unmixing (Correlative 3D-Confocal/2D-dSTORM: 450–500 nm for α-Tubulin-Atto425 and 510–540 nm for γH2A.X-Alexa Fluor488; Correlative 3D-Confocal/TIRF: 410–480 nm for Hoechst 33342, 450–500 nm for Phalloidin-Atto425, 490–550 nm for 53BP1-Alexa Fluor488, 580–620 nm for Pin1-Cy3, and 650–720 nm for α-Tubulin-Alexa Fluor647). Z-Stacks were acquired by moving the sample with a piezoelectric Z stage, in galvanometric scanning mode.

#### 2.5.3. SIM Microscopy

All the measurements were performed by the Nikon N-SIM Super-Resolution module using a 1.49 NA 100× objective. Five phases of the sine wave pattern are recorded at each Z position, allowing the shifted components to be separated and returned to their proper location in frequency space. Three image stacks are recorded with the diffraction grating block 3D EX V-R 100×/1.49, sequentially rotated into three positions 60° apart, resulting in nearly rotationally symmetric support over a larger region of frequency space. Laser power and camera exposure times were adjusted to achieve the optimal balancing between high dynamic range and photobleaching. Excitation was provided by a LU-NV laser unit (405, 488, 561, 647 nm lasers).

#### 2.5.4. TIRF Microscopy

TIRF Images were acquired by a Nikon CFI SR Apochromat TIRF 100× oil objective (1.49 NA), configured for total internal reflection illumination. Image format was set to 1024 × 1024 pixels, with a pixel size of ~0.065 μm, to optimize the homogeneity of illumination in total internal reflection conditions.

#### 2.5.5. dSTORM Microscopy

Single-molecule imaging was performed with a super-resolution Nikon N-STORM microscope configured for either total internal reflection fluorescence (TIRF) or oblique incidence excitation, employing a continuous activation mode (both the activator and imaging lasers were continuously on). Alexa Fluor 647 and Cy3 dyes were excited by using 561 nm and 647 nm laser wavelengths, while a 405 nm solid-state laser was used for the activation of dyes. A multi-band dichroic mirror (C-NSTORM QUAD 405/488/561/647 FILTER SET; Chroma Technology, Bellows Falls, VT, USA), combined with 561 nm- and 647 nm- filter cubes (IDEX Health & Science, Semrock, Rochester, NY, USA), was used to filter the fluorescence excitation. Acquisitions of the two channels were performed in sequence. The 488 nm channel was employed for the fiducial marker acquisition (Fluorescent Nanodiamonds with Streptavidin, 40 nm, Adamas Nanotechnologies), 1 every 1000 frames, required to correct the drift between frames and the spatial shift between channels. The fluorescence emission of all channels was collected using a Nikon CFI SR Apochromat TIRF 100× oil objective (1.49 NA) and finally detected by the digital CMOS camera. The number of frames and exposure time per channel depend on the density pattern of the immunostaining and the dye blinking state (15.000 frames in continuous mode at 20 ms of exposure time per channel). Z drift was minimized thanks to a hardware autofocusing system (Perfect Focus System (PFS) (Nikon instruments, Tokyo, Japan). Under constant illumination and buffer conditions, the dyes typically started in the fluorescent state, switched to a dark state, and spontaneously recovered to a fluorescent state several times before photobleaching. The power of both activation and excitation lasers is dependent on the blinking efficiency of the single fluorophores. The modulation of lasers was adjusted with neutral density filters to reduce the photobleaching effect during the preliminary observation for the optimization of the acquisition parameters.

##### dSTORM Imaging Buffer 

Single-molecule dSTORM imaging for each dye was performed in an imaging buffer that included 10 mM Tris (pH 8.0), 50 mM NaCl, 10% Glucose, an oxygen-scavenging system “GLOX” (5.6 mg/mL Glucose Oxidase, 0.34 mg/mL Catalase), and 100 mM Mercaptoethylamine (MEA; Stock 1 M (77 mg MEA + 1.0 mL 0.25 N HCl)). The solution was kept at −20 °C and used within 2–3 weeks from preparation.

##### Drift Correction Marker

Nitrogen-vacancy-center fluorescent nanodiamonds with streptavidin (FNDs; 40 nm; Adamas Nanotechnologies, Raleigh, NC, USA) were employed as markers for dSTORM. The stock solution (in FBS with 0.1% BSA at 1 mg/mL (1% *w*/*v*)) was diluted 1:50 in DPBS 1X and added to the cells overnight at 4 °C.

### 2.6. dSTORM Image Reconstruction and Colocalization Analysis

Single Molecule Localization fitting was performed with Offline N-STORM Analysis module (NIS Elements software, version 5.42.01, Laboratory Imaging s.r.o, Praha, Czech Republic) considering drift and chromatic aberrations. Before exporting STORM images, the Gaussian size of the single localizations and the format of the reconstructed images were set to optimize correlative imaging. In the super-resolution image reconstruction, each molecule is represented by a Gaussian spot localized by the centroid position, with the localization precision obtained from the single-molecule fit, and by an amplitude value proportional to the number of emitted photons. We finally generated a dual-color STORM image of 41 × 41 μm (10 nm/px) with a Gaussian size of 10 nm. Colocalization analysis was performed according to an Image Cross-Correlation Spectroscopy pipeline [35]. The object-based quantification of the events described in the text was instead performed by the A.M.I.CO software detecting 53BP1 and DynLL1 spatial clusters of 50 nm size. In correlative confocal-STORM imaging a mask was obtained by segmenting the additional confocal channel. The relative distance distributions were computed thanks to the procedure coded in the software.

### 2.7. Automated Acquisition Protocol

The acquisition protocol was coded into the Microscope Control Software by the JOBs plugin module for the execution of the following steps:Image acquisition parameters were optimized for every channel before launching the automated acquisition routine of the control software.A mosaic image of the entire dish in the DNA-dye channel was first acquired by a 4×/N.A. 0.13 objective, and the resulting image was used to locate regions of interest (ROIs) of fixed dimension (1.4 × 1.4 mm) to then be acquired. This way, regions with optimal cell density and free of defects (e.g., dust, antibody precipitates, and large cell aggregates) could be excluded from the acquisition.A position list of the centers of the selected ROIs was created and the objective was switched to high-resolution imaging (60× N.A. 1.4 Oil-Immersion)For every region, a two-step autofocus procedure was employed based on the DNA-dye signal. The microscope stage was first moved to the center of each ROI and used to perform a wide-range search (from 15 to 30 microns) with the optimal focus. Then, a series of regularly spaced points were selected to fit the position of the focal plane over the entire region by searching the best focus on a more restricted range. This procedure simultaneously optimizes the quality of the acquired images and the time spent on the autofocus search.The entire ROI was acquired by collecting adjacent positions. Every channel was acquired sequentially.The same procedure was applied to each selected ROI.The stored images were then analyzed by the developed A.M.I.CO analysis software: XY cell location was retrieved from selected subpopulations of interest, converting the image coordinates into stage positions and passing the list to the microscope control software. The targeted cells could then be re-located and re-acquired with high- and super-resolution imaging methods for further analysis.

### 2.8. A.M.I.CO Image Analysis

MCF10A cell samples was analyzed by the A.M.I.CO analysis package developed within the open-source ImageJ platform using its macro programming language. 

The package is described in detail in a previous work [24]. The software is freely available upon request or it can be downloaded from the GitHub public repository (https://github.com/MarioFaretta/AMICO, accessed on 20 November 2022). Since it is not possible to code to all the information required (e.g. format of the position lists for the microscope, format of the acquired images and metadata) by different microscope brands, a customization step is required to the users for adapting the code to their set-ups. We are available to provide help for these modifications.

Briefly, A.M.I.CO is composed of two separate modules. The first module (A.M.I.CO_Union) is dedicated to data browsing, parameter setting, and image analysis. Multi-parameter image analysis requires a pre-processing procedure to subtract the background (i.e., the Rolling-Ball algorithm) and to homogenize illumination (i.e., flat-field correction). After setting the proper segmentation conditions for single cell identification, images were processed, and a single-cell data repository was created containing, in each row, single-cell measurements. Geometrical parameters (area, circularity, and physical XY coordinates in the field of view) were calculated for each identified cell. The fluorescence mean value per pixel was also computed, together with the integrated fluorescence intensity per cell. For every analyzed channel, sub-compartments could also be defined for the measurement of intracellular details (e.g., spots, and nucleoli). Physical and fluorescence parameters were re-calculated for each detected intracellular event. The analysis procedure stores data in a tab-delimited text file containing the calculated measurements, which are classified into cell- or sub-compartment-related values.

The second data-analysis module (A.M.I.CO_Plotting) presents an interface similar to flow cytometry analysis programs. Histograms and 2D multicolor dot plots were generated to represent all the events in the sample, allowing the definition of specific regions for statistical measurements and logical gates for selective targeting of events. With the integrated cell-retrieval algorithm, every event (cell or subpopulation) can be visualized and physically re-positioned on the microscope stage, for further acquisition with different imaging modalities and analysis.

### 2.9. Correlative Microscopy

As a first step, a widefield image, centered on the cell of interest, was acquired in the channel employed for both STORM and Confocal imaging. Spatial sampling was set to satisfy the Nyquist criterion (65 nm pixel size, no binning). The digital size of the collected image was adjusted to obtain a similar field of view (FOV) between WF and STORM images, but with a different spatial sampling (65 nm for WF, 160 nm for STORM). STORM images were in fact acquired by inserting a 0.4X Relay lens to enlarge the FOV and to increase the number of collected photons per pixel. A digital size of 256 × 256 pixel image in a STORM image matches with 620 × 620 pixel of WF ROI (65 nm/pixel). WF images were also used to check the potential drift in the pictures collected by the STORM acquisition and the confocal image.

For 3D-confocal imaging, the pixel size was adjusted in order to match the spatial sampling of the WF image (about 65 nm). The pinhole size was set to 0.8 Airy unit. Bidirectional galvanometric scanning was enabled and the confocal acquisition parameters, such as line average, pixel dwell time, laser power, detector filters, and gain were adjusted to increase the quality signal without saturation while also minimizing background signal and photobleaching. A Z-stack with 200 nm spacing was finally acquired covering the entire cell volume.

As next step, the 2-color STORM acquisition was run. STORM Analysis on the two acquired datasets identifies the positions of the fluorescent reporter molecules to reconstruct the two STORM images, including a correction of stage drift during the acquisition. Final alignment of the WF, confocal, and STORM images was performed after acquisition. The confocal and high-resolution WF images were aligned using registration software (both the plugin included in the NIS software and the open-source Turboreg plugin in the ImageJ platform were employed in this work). Reference points were based on targeted biological structures (e.g., DDR foci, microtubules) identified in the two imaging modalities. The calculated shift was then employed to align the drifted widefield image, acquired with the 0.4X Relay lens of the STORM acquisition module. The registration parameters were finally applied to obtain the aligned reconstructed STORM images (see Appendix A).

## 3. Results and Discussion

### 3.1. Application of Multi-Modal Correlative Pipeline Combining Imaging Modalities with Scalable Spatial Resolution

As previously mentioned, the optical microscope is the point of convergence of many techniques providing the optimization of different “resolutions”, i.e., (3D) spatial resolution, temporal resolution, sensitivity, optical penetrance, statistical sampling, and content (number of simultaneously measured parameters) (Figure 1).

For example, super-resolution microscopy, i.e., STED, STORM, or MinFLUX, allows the observation of molecular events with a resolution of less than 50 nm and a molecular localization precision that can reach the sub-nanometer level [38]. However, nanoscopy suffers from a limitation in the number of simultaneously detectable parameters, of a relative slowness in the acquisition time, and of a reduced statistical sampling, in terms of samples throughput. Standard confocal microscopy instead maintains optimal performances in 3D reconstruction at the expense of a limited speed. These limitations have been overcome by the parallelization in the excitation beam in spinning disk microscopy that unfortunately keeps, as the original point-scanning microscope, a drop in the sensitivity due to the presence of the emission pinholes requested to achieve optical sectioning. These are only a few examples of how multi-modal microscopy can provide a way to remove these obstacles by the simultaneous employment of the different microscopy technologies according to the required piece of information, to gain the best performances from each of them.

We developed and present here a pipeline (Figure 2) for correlative optical microscopy based on two main concepts: (i) re-scalable resolution and (ii) analysis-driven image collection.

According to the step executed in an imaging process, the real requested resolution can span from the order of microns, to identify and classify single cells, to the super-resolution range below 100 nanometers. As traditionally performed in correlative optical-electron microscopy, one target is the localization of selected cells and/or structures to analyze them at the highest resolution.

In the presented protocol, this task is accomplished in a more exhaustive definition with the event-targeting based on the selection of complex phenotypes evaluated according to an image-cytometry approach. This way, it is possible to obtain (i) a high statistical sampling able to collect and analyze several thousands of cells, also making it feasible to target poorly represented cell populations; (ii) a classification of the cells based on complex phenotypes obtained by quantitative measurements of the fluorescence signals on several fluorescence channels with high sensitivity; (iii) an indirect temporal localization (e.g., cell cycle localization); and (iv) a quantitative diffraction-limited analysis and targeting of intracellular structures [24,34].

This analysis-driven phenotype selection provides the advantage of a more precise, faster, and easier sample relocation granted by the use of only one microscope, instead of separate imaging platforms. The operational flow concludes with the physical repositioning of the microscope stage on the target event, allowing the execution of a second image-collection phase: thanks to the combined use of the different high- and super-resolution techniques present in a hybrid setup, the rescaling of the spatial resolution to a sub-diffractive level is achieved for a deeper investigation of the molecular mechanisms. According to the addressed question and the requested final resolution, the developed pipeline allows performing the following stepsd on both targeted cells and subcellular structures (Figure 2): (i) a 3D diffraction limited confocal analysis; (ii) a sub-diffraction resolution (120 nm) 3D SIM analysis; (iii) a correlative microscopy Confocal-STORM analysis that combines the sequential imaging of the field of view by 3D-confocal microscopy (to visualize immunolabeling in the whole cell volume) and by 2D-dSTORM (to visualize protein distribution at the nanoscale in a selected plane); and (iv) a combined Confocal-TIRF-STORM imaging for the high-resolution analysis of the cell interface.

Moreover, the automation of the process led to a dramatic increase in the throughput of the high-resolution analysis, granting an optimization in the timing requested by the experiment and in the storage space for the final collected data.

### 3.2. Application of a Multi-Modal Correlative Pipeline, Step 1: High Statistical Imaging Acquisition and Analysis by Image Cytometry

To validate the potential of the pipeline, we focused on the 53BP1 protein, a member of the DNA Damage Response (DDR) machinery. Several pieces of evidence showed a large panel of interactors revealing novel putative functions that go beyond 53BP1’s role in damage recognition and DNA repair [39,40,41,42,43,44,45]. Recent publications [36] showed how 53BP1 localization can modulate a functional switch between a classical involvement in DDR and support to p53 activity by moving from DNA Damage foci to the nucleoplasm. Consequently, it becomes of paramount importance to shed light on the structure–function relationships of this protein and its molecular partners, associating a precise measurement of the number of molecules in the cell to their intracellular localization and understanding how these parameters can vary according to the cell state (e.g., cell cycle localization).

Among 53BP1 molecular interactors, we selected the Dynein Light Chain DynLL1/LC8, a molecular hub promoting dimerization of a broad range of targets that is essential for the integrity of many cellular subsystems, e.g., Myosin, Dynein, and apoptotic factors [45,46]. DynLL1 is a 53BP1 effector that is recruited to DD foci [47,48].

Cells’ positions in the cell cycle were estimated by analyzing images acquired by a 1.4 NA 60× oil-immersion objective to guarantee the maximal resolution and sensitivity [24]. Two different experimental approaches were adopted (Appendix A). The first one employed an EdU pulse to mark actively DNA-replicating cells combined with DNA content quantification, measured by the DNA dye total intensity per cell (e.g., DAPI, Hoechst, and Chromomycin). When the number of detected molecular markers must be increased, a second protocol was followed: the cell cycle position was estimated according to the DNA content only, making the EdU channel available for additional signals. Mitotic cells can be distinguished based on the DNA signal (i.e., integrated intensity for the DNA content and mean intensity for DNA condensation) and morphological parameters (i.e., size (area) and shape (circularity)). When available, indirect mitotic markers can be combined with these morphological parameters. Phosphorylated histone H2A.X (γH2A.X), entrapped in the condensed chromatin, lights up mitotic chromosomes, providing an additional distinctive phenotype of mitosis by high mean intensity (Appendix A). Alternatively, the proliferation marker KI67 accumulates on the surface of chromosomes, again causing a marked increase in the mean intensity [24]. This way, all the cells accumulated in the acquired images were classified according to their cell-cycle stage (Appendix A).

A Proximity Ligation Assay (PLA) was employed to highlight and localize the sites of the 53BP1-DynLL1 interactions. PLA detection, thanks to the wide spectral range of the widefield microscope, was coupled to the staining of other eight antigens (Figure 3a) to delineate intracellular structures or to provide evidence of functional cell states: (i) DNA, for cell cycle localization and nucleus marking; (ii) Actin, to delineate cell membranes particularly during the mitotic phase; (iii) KI67; and (iv) Lamin A to mark proliferation activity (KI67 was also employed as a mitotic marker (together with γH2A.X), while Lamin A was useful to detect prophase and telophase onset); (v) γH2A.X was used to quantify and localize DNA damage; (vi) Tubulin for the spindle visualization; (vii) 53BP1; and (viii) DynLL1, to measure their expression profile and localization during the cell cycle.

The developed image-cytometry tools provided a precise classification of cell-cycle-related phenotypes, generating statistics of the corresponding number of events, global, or relative (restricted to a gated subpopulation), percentage representation, and the average value per cell for every measured parameter in the selected population.

The distribution of 53BP1-DynLL1 sites of interactions was consequently (i) quantified at high sensitivity; (ii) “temporally localized” during the cell cycle, demonstrating that it is not phase-specific; (iii) spatially localized in cell compartments, detecting its presence in both nuclear and cytoplasmic space; and (iv) targeted in intracellular structures, e.g., γH2A.X foci. The statistical sampling reached is in the order of thousands of cells (n = 5156) (Figure 3b), with the simultaneous optimization of signal-to-noise ratio and spatial resolution (diffraction limited at NA 1.4).

Each cell can be physically relocated on the microscope stage, correlating its file of origin to the spatial coordinates generated by the employed acquisition software. The A.M.I.CO analysis macro can read the calibration and acquisition parameters available in the data-repository specific to the microscope and recalculate the stage coordinates for selected events (Figure 2). The presented procedure thus provides an example of how high-content analysis can be originated and developed from the correlative pipeline.

### 3.3. Application of a Multi-Modal Correlative Pipeline, step 2: Event Targeting for Confocal and SIM Microscopy to Achieve 3D High-Resolution Imaging

53BP1-DynLL1 interaction was detected throughout the whole cell cycle. 53BP1 localization extended out of the nucleus in the cytoplasmic space, particularly during the mitotic phase (Figure 3). Interestingly, the literature has shown the involvement of both the proteins in the mitotic spindle-checkpoint activation, with 53BP1 being excluded from the condensed chromosomes but recruited to the centrosome and spindle pole [49,50], while DynLL1 has a specific function in spindle orientation control [51].

An attempt to focus on the relative localization of the two proteins during mitosis revealed the inefficacy of 2D widefield fluorescence imaging due to the increase and spatial remodeling of the cell volume during the division process.

Thanks to the above-mentioned possibility to identify all the cell-cycle stages and to physical reposition target events on the microscope stage, mitotic cells were analyzed by diffraction-limited confocal microscopy (Figure 4a).

The collected Z-stacks were thus reordered to reconstruct the temporal evolution of mitosis in 3D. Of note, once the confocal-imaging parameters were set, the entire process was completely automatized without any user intervention. Considering acquisition parameters satisfying Nyquist sampling in the three spatial dimensions (XY pixel set to 60 nm and Z sampling to 125 nm) and an eight-channel sequential acquisition, an average duration of 7 min per cell was registered with a speed of around 1.5 frames per second per channel. Acquiring 30 cells (eight-channel Z-stacks) required about 3.5 h. High-content 3D analysis revealed that some 53BP1-DynLL1 complexes resided on tubulin filaments on the mitotic spindle during mitosis highlighting the above-mentioned functional role of the two proteins (Figure 4b).

Image-Cytometry High-resolution widefield analysis showed 53BP1 and DynLL1 proteins in the interphase cells were diffused in the nucleoplasm and accumulated in DNA damage foci as marked by simultaneous γH2A.X accumulation (Figure 3a). Among the damaged sites, of particular interest is a class of large 53BP1 clusters: some DNA damages, formed during the cell cycle, do not activate the checkpoints to arrest its progression. They are instead equally partitioned among the duplicated genome. However, after cell division, at the beginning of the new cycle, daughter cells recognize the inherited loss of DNA integrity and fix the problem during the G1 phase before starting the new replication [52,53].

Thanks to the developed computational tools, morphological and fluorescence analysis can be extended not only to cells but also to sub-compartments, considered as independent entities. Among the 53BP1 foci populations, it was consequently possible to isolate the ones that are clearly distinguishable by intensity and size. Linking back these foci to the cells of origin evidenced a G1-specific distribution that can be physically retrieved on the microscope (Figure 5a).

However, the high molecular crowding detected in these large spots suggested a first scale-up in the spatial resolution. We thus employed the correlative pipeline to switch to 3D-SIM microscopy, which is able to resolve structures up to 100–120 nm on multiple fluorescence channels (Figure 5 panel b). At least in our set-up, which presented an important time limit due to the mechanical movements employed to generate the spatially shifted masks, SIM-imaging collection is a highly time-consuming process. One of the advantages of the analysis-driven acquisition consisted of the restriction of the field of view to the cell of interest: the collection of the 15 images per channel per plane, required for the enrichment in the high spatial frequencies, can be consequently sped up. Moreover, as previously mentioned for the high-resolution 3D-confocal imaging, image collection did not require any user intervention and proceeded autonomously for several hours after the set-up of the acquisition parameters.

SIM imaging suggested that the G1-specific large clusters were partitioned in resolution-limited substructures (120 nm of size) containing both 53BP1 and DynLL1 as further confirmed by the presence of PLA spots.

### 3.4. Application of a Multi-Modal Correlative Pipeline, Step 3: Combined 3D-Confocal/2D-dSTORM Microscopy for Multicolor 3D Localized Imaging to Target and Super-Resolve Intracellular Structures

PLA localization far from chromatin compartment was not limited to mitosis, also being visible in the cytoplasm of interphase cells (Figure 3 and Figure 5). However, despite its sensitivity in detecting proximity in the 50 nm range, PLA may present important limitations due to the low ligation and amplification efficiency, which underestimates the number of real interactions [54,55,56]. The increased spatial resolution given by dSTORM microscopy enables a more detailed analysis of whether two proteins co-localize at the single-molecule level, obtaining information about their spatial distribution and estimating the fraction of involved molecules. Unfortunately, the restricted spectral range covered by fluorochromes exhibiting the blinking properties required by dSTORM made multicolor imaging quite a hard task, usually limiting it to two channels. Furthermore, an accurate dSTORM 3D analysis extended over the whole cell range is challenging and limited to a section of about 1 micron, making it infeasible to obtain a real reconstruction to navigate in the intracellular space. Coupling to confocal microscopy can thus increase the number of detectable parameters with a resolution in the three dimensions that simultaneously allows targeting single-molecule localization microscopy towards selected regions of interest.

We thus implemented a correlative microscopy procedure to reconstruct a diffraction-limited, optically sectioned volume of a target cell to identify and position a 2D-dSTORM image plane where proteins were localized at the single-molecule level (Figure 2). The signal-to-noise ratio of three-dimensional confocal imaging was improved by post-processing image deconvolution.

The nanoscale distribution of 53BP1 and DynLL1 was thus reconstructed. In parallel, phosphorylated histone γH2A.X and α-Tubulin were imaged by confocal microscopy to delineate the cell compartments where PLA previously localized the formation of some putative complexes. The accumulation of both 53BP1 and DynLL1 in smaller clusters was evident inside DD foci localized by γH2A.X (Figure 6a). At the same time, single-molecule localizations were easily recognizable on the cytoskeleton in interphase cells, confirming the presence of the two partners. To increase resolution on the formation of a 53BP1-DynLL1 putative complex associated with the spindle checkpoint, mitotic cells were selected according to the procedure described above. Figure 6b, shows how 53BP1 and DynLL1 molecules spread over the entire cytoplasm residing both on the microtubule network and at the chromosome periphery from prophase to telophase.

### 3.5. Application of a Multi-Modal Correlative Pipeline, Step 4: Combined Confocal-TIRF Microscopy

The data collected by the multi-modal microscopy pipeline suggest that 53BP1 molecules can be localized in the nuclear compartment, in and out the DDR foci, and in the cytoplasm, simultaneously on the tubulin network and diffused throughout the whole intracellular space. However, an overview of the acquired images inevitably leads to the question of whether 53BP1 proteins may reach the plasma membrane.

Interestingly, 53BP1 was originally identified as a binder of wild-type but not mutant p53 together with another protein named 53BP2. 53BP2 shows a cytoplasmic localization [57], and some of its variants interact with RAS and Insulin membrane receptors [58]. In MCF10A cells, 53BP2 accumulated in large clusters on the contact region of the plasma membrane of adjacent cells (Figure 7). We thus employed these 53BP2 structures to further validate the ability of the developed analysis software in isolating selected targets and repositioning them on the microscope stage. The high intensity and elongated shape of the 53BP2 regions successfully identified them in the collected images (Figure 7a). Widefield microscopy was then correlated to the dSTORM single-molecule localization map, showing that 53BP1 was also present in the targeted structures and, as a consequence, on the plasma membrane (Figure 7b).

To fully address the question of the putative 53BP1 localization in the plasma membrane compartment, we finally inserted in the correlative pipeline total internal reflection microscopy (TIRF), which provides one of the highest Z resolutions among the far-field optical technologies [59]. Cells were stained to visualize DNA, Phalloidin, α-Tubulin, DynLL1, 53BP1, assigned to the different cell-cycle phases according to the procedure described above, and repositioned on the microscope stage for combined TIRF-Confocal microscopy. The TIRF evanescent field excites fluorescence with an exponential decay in power that confines photon emission within a distance of 100–120 nm from the cell–glass interface [60]. As a result, the increased signal-to-noise ratio and the dramatic reduction in the molecular crowding led to an increased ability to distinguish molecule distribution. Considering that the thickness of a confocal slice is in the order of the micron, TIRF provides an improved view with respect not only to widefield but also to confocal microscopy (Appendix A). Even if TIRF cannot reach the single-molecule localization precision granted by dSTORM, it essentially provides an extremely good real resolution at the molecular level, a higher number of simultaneously detectable fluorescence signals, and a time resolution that is order of magnitudes higher than the one typical of single-molecule localization microscopy. Recognition of the same structures in confocal and TIRF optical plane made the alignment process, requested by correlative analysis, particularly straightforward.

As was previously used, correlative imaging with confocal microscopy was applied to reconstruct the 3D spatial distribution of the proteins of interest. The presence of co-localizing and proximal 53BP1 and DynLL1 molecules, both on and far from the tubulin network, confirmed that the interaction of the two targeted proteins may occur in several compartments of the intracellular space (Figure 8).

### 3.6. Application of a Multi-Modal Correlative Pipeline, Step 5: Interpreting Data with Optimized Analysis Approaches

A dynamic modulation of the spatial resolution in the collected data requires targeted analysis procedures able to extrapolate all the information obtained by the combination of different imaging modalities.

Besides the large statistical sampling reached by the automated collection of thousands of cells and the possibility of targeting selected fractions of events (e.g., cell-cycle phase distribution), the extension of the number of simultaneously acquired parameters by multi-modal correlative microscopy allows localizing signals up to the single molecule scale (e.g., by STORM and TIRF microscopy) in specific cell compartments (e.g., by Widefield and Confocal microscopy).

A single-molecule localization analysis can complement the classical biochemical assays of biomolecular interactions provided by, for example, PLA. As an enzyme-based reaction, PLA has limited efficiency. Moreover, the size of DNA rolling circles and the high number of fluorescently labeled nucleotides heavily limit the dynamic range for the digital quantification of targeted molecules [61]. More detailed information on the real extent of the interaction among two molecular species and its stoichiometry can be consequently derived by correlation analysis of single-molecule data.

We employed targeted data-analysis procedures to quantify the 53BP1-DynLL1 interaction shown in the previous paragraphs. 53BP1-DynLL1 PLA analysis showed (Figure 3, Figure 4 and Figure 5) how the two proteins can be engaged in macromolecular complexes throughout the whole cycle and outside the nucleus. DynLL1 is a protein hub associated with the cytoskeleton. Considering the previously reported data on the link between the cytoskeletal proteins and DDR response [62,63,64] and the above-mentioned role in mitosis regulation, we addressed the question of whether the 53BP1-DynLL1 could be localized on microtubules and to which extent.

The low PLA efficiency limited the detection of putative complexes in highly enriched and dense molecular compartments, and even if PLA showed cytoplasmic signals, the reduced number of spots is not sufficient for a statistically significant analysis of their cytoskeletal compartmentalization. The correlative confocal-STORM data (Figure 9a) allowed the measurement of the cytoplasmic co-localization of 53BP1-DynLL1. The fraction of putative complexes adjacent to the cytoskeleton was simultaneously isolated thanks to the localization of beta-tubulin by confocal microscopy. ICCS analysis of clustered (50 nm) single-molecule events [65] revealed that the 53BP1-DynLL1 proximity was equally partitioned between cytoplasm and microtubules. This profile was maintained throughout the entire cell cycle, since the deviations registered during the different phases of the mitosis were not statistically significant.

ICCS analysis presented the advantages of a pixel-based analysis, e.g., speed, that can be particularly useful with a high number of analyzed images, but limiting and difficult to interpret when looking at frequency and stoichiometry of interactions. Object-based procedures can be more effective and easily interpreted.

To obtain a precise quantification of the putative interaction, clustered (50 nm) STORM images were segmented, employing the A.M.I.CO software in the two channels and in the different phases of mitosis (Figure 9b). A mask of the microtubule network was simultaneously created by the registered confocal data. Clusters separated by a relative distance of 50 nm were considered adjacent. The analysis confirmed a mitosis-stage-independent interaction with a 1-to-1 ratio (between 90–98% for the 53BP1 and 87–91% for the DynLL1 colocalizing fractions) between the clusters of the two molecular species.

Finally, Figure 9c, reports an object-based analysis of the 53BP1-DynLL1 interaction on TIRF-collected images to analyze the relative distribution of the targeted protein on the cell membrane. The two channels were segmented and reciprocal distances of the recognized objects were calculated. An arbitrary value of about 250 nm was considered as the limit of adjacency. Results evidenced a higher percentage of proximally located molecules on the basal membrane (about 30–40%) also influenced by the lower number of molecules detected in the compartment.

## 4. Conclusions

The modern optical fluorescence microscope is a hub hosting several novel technologies with different performances. The wide heterogeneity of the samples and of the addressed experimental questions leads to the need to choose the best performer from time to time. Multi-modal microscopy allows a continuous switch among multiple imaging modalities by registering data from each source. We presented here a set of tools for a pipeline designed to correlate different optical technologies that is characterized by a dynamic adaptation of the spatial resolution associated with an extremely high statistical sampling, giving robustness to the data. The procedure exploits automation to increase and speed up the image collection, and it also includes feedback between the acquisition and analysis process, which allows the intelligent selection of the targets according to the real experimental question and makes feasible the multi-modal correlative microscopy approach. The final image analysis procedures must obviously take into account the dynamic rescaling of the spatial resolution and the deriving limits in inferring conclusive results. We concluded the description of the experimental pipeline by reporting an example of data analysis. High statistical sampling was employed to obtain temporal, i.e., cell-cycle, localization, while the variable resolution images in the multi-modal microscopy acquisition allowed targeting, at lower resolution, an intracellular compartment to evaluate there, at the highest spatial resolution, the reciprocal distribution of molecular species.

The presented data just provide a demonstration of an experimental workflow focused on a selected biological target, i.e., the interaction between 53BP1 and DynLL1. However, the reported observations require additional experimental confirmations before considering them as definitive data for the characterization of the structure–function relationships between the two proteins. They just represent an example of how multi-modal microscopy can provide solutions allowing the travel inside complex phenotypes and temporal and spatial intracellular localization with variable and optimized resolutions.

Future developments will obviously aim at strengthening the concept of the “intelligent microscope”, i.e., analysis-driven acquisition, due to the contribution of artificial intelligence to increase the ability to process analytical problems generated by the complexity of the biological world.

## Figures and Tables

**Figure 1 cells-12-00354-f001:**
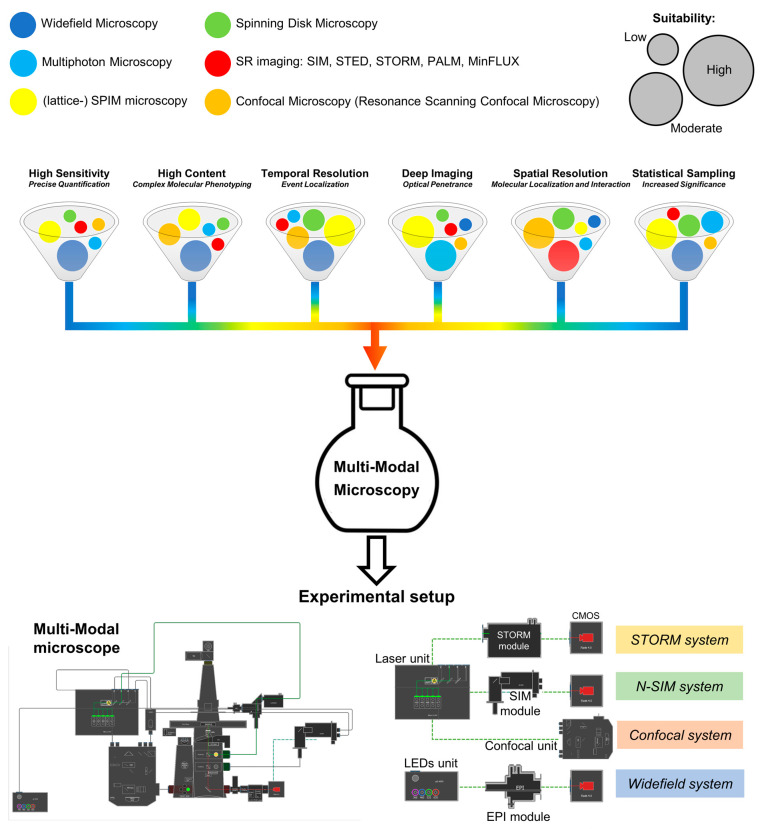
Correlative Multi-Modal Microscopy. Every fluorescence microscopy technique shows different performances (low, moderate, high) according to the considered resolution (e.g., spatial, temporal, statistical sampling). Multi-Modal Microscopy can overcome these limitations by combining the different technologies as shown in the set-up employed in this work.

**Figure 2 cells-12-00354-f002:**
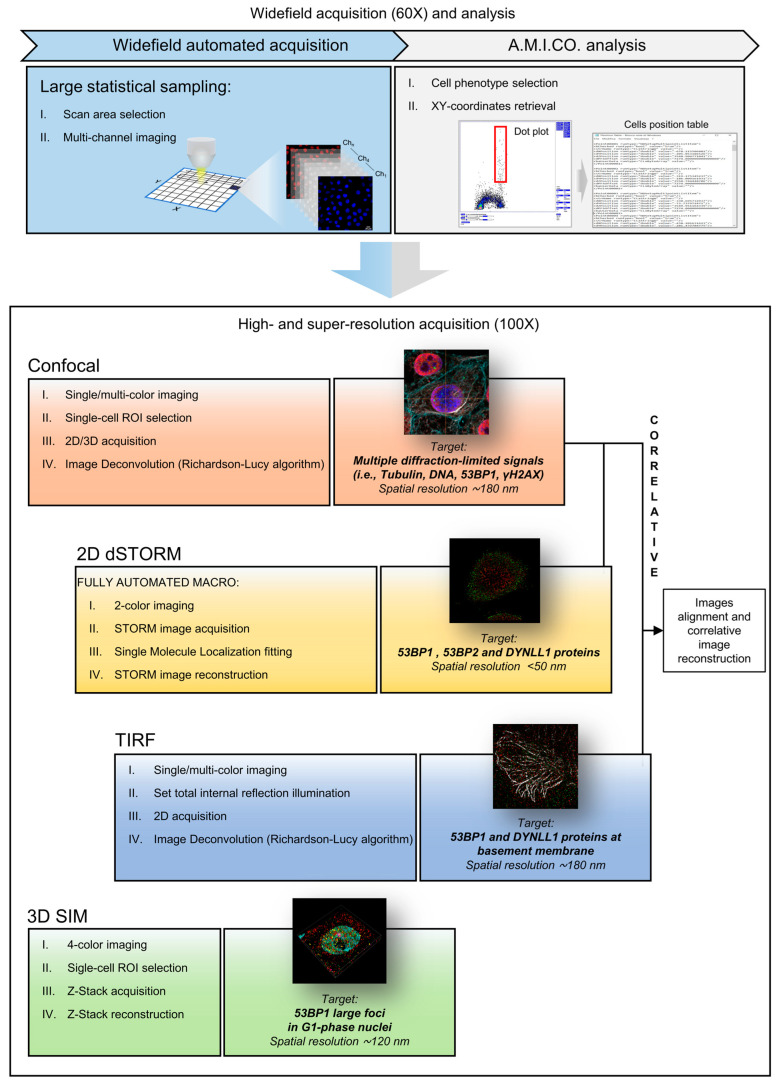
Multi-Modal Multi-Resolution Workflow for Correlative Experiments. First, images are collected by an automated fluorescence-microscopy workstation following an analysis-driven protocol (A.M.I.CO analysis) that allows the automated acquisition of multi-channel images with high resolution and their processing to identify single events. Cell targets of interest (specific cell phenotypes) are thus selected and physically retrieved on the microscope by converting the image coordinates into stage positions. High- and super-resolution imaging (including confocal, STORM, TIRF, and related correlative modalities) is then performed on the targeted regions.

**Figure 3 cells-12-00354-f003:**
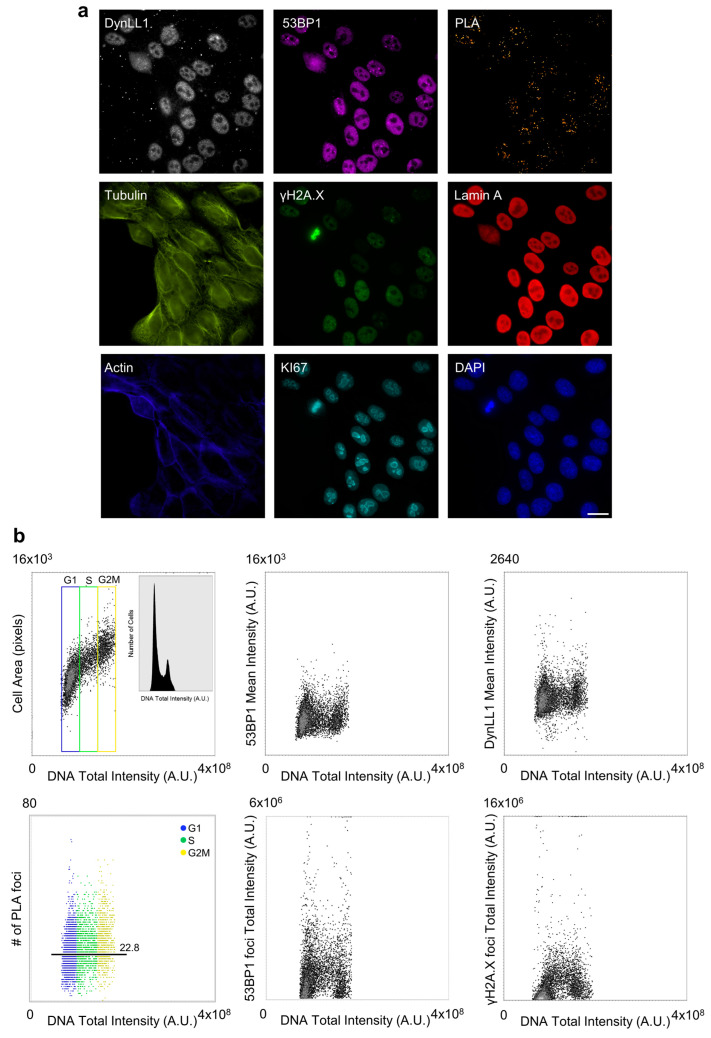
Image cytometry analysis of protein expression and protein–protein interaction (detected by Proximity Ligation Assay (PLA)) during the cell cycle. (**a**) Images of exponentially growing MCF10A cells were collected with a 60× 1.4 Oil Immersion Objective according to the described automated widefield-acquisition procedure. Samples were stained for the detection of DynLL1, 53BP1, 53BP1-DynLL1 PLA, Tubulin, γH2A.X, Lamin A, Actin, KI67, and DNA (DAPI). Scale bar: 20 µm. (**b**) The dot plots report the bivariate distribution of DNA content (*X*-axis) versus (*Y*-axis) (i) cell area and number of cells (n = 5156) (histogram in the box) to detect the cell-cycle distribution according to the DNA content—G1 (blue), S (green), and G2M (yellow); (ii) mean intensity per pixel per cell of 53BP1 and DynLL1 to estimate the protein content during the cell cycle; (iii) total intensity of 53BP1 and γH2A.X foci to measure the DNA Damage in relation to the cell-cycle position; and (iv) number of interaction spots detected by a PLA assay between 53BP1 and DynLL1 during the cell cycle.

**Figure 4 cells-12-00354-f004:**
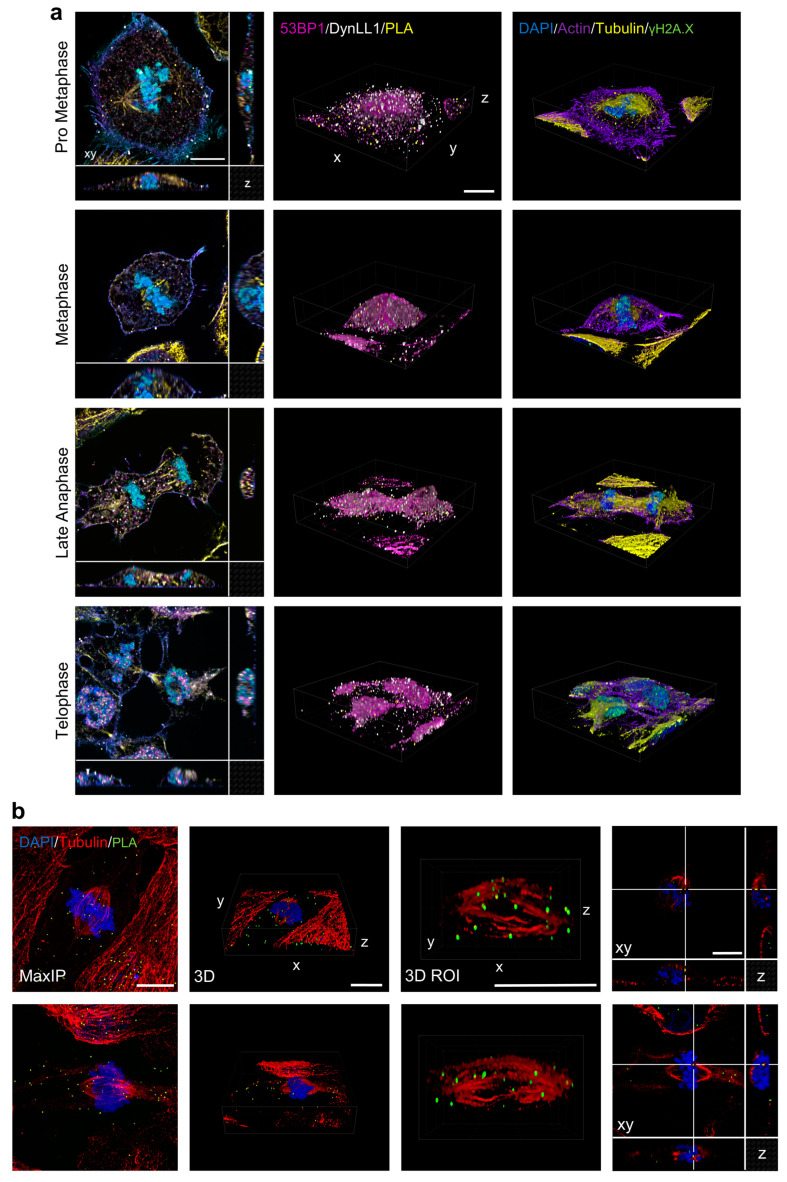
Analysis-Driven Confocal Acquisition of Targeted Mitotic Cells. (**a**) Representative Confocal Stacks of cells in the different stages of mitosis, selected according to the procedure described in the text. The first column shows multiple views, along the optical axis (XY) and lateral planes (XZ, right; YZ, bottom). Three-dimensional volume projections are shown in the second (PLA, 53BP1, DynLL1) and third (DAPI, Actin, Tubulin, γH2A.X) columns, respectively. (**b**) Representative 3D volume projections and lateral planes’ reconstruction (last column) of PLA, Tubulin, and DAPI signal shows the association of the PLA- interaction spots with the cytoskeleton. Scale bar: 10 µm.

**Figure 5 cells-12-00354-f005:**
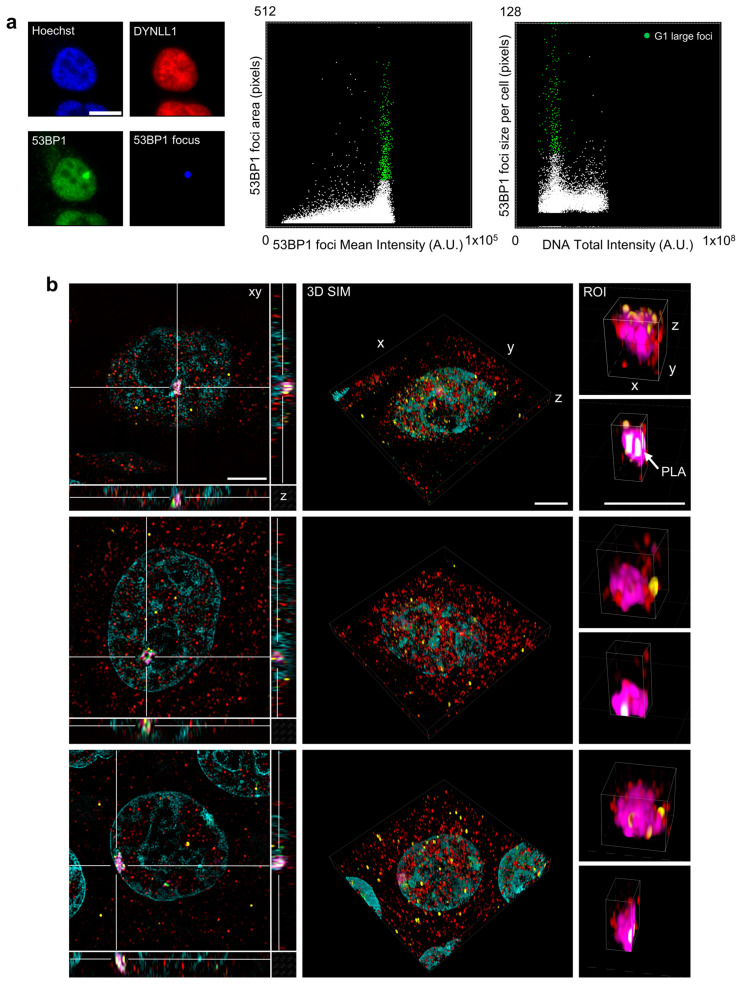
53BP1 large foci targeted by Image-Cytometry Analysis. (**a**) 53BP1 shows accumulation in large foci that can be identified by high mean intensity and area when considering the distribution of the spots independently from the cell of origin (dot plot in the middle column). When reassigned to the originating cells, the targeted cell population localizes in the G1 phase of the cell cycle (dot plot on the right) identified also by the high average size of the 53BP1 foci. (**b**) Lateral-plane reconstruction (first column) and 3D volume projections of cells (second column) and large 53BP1 foci (third column) of representative events from the previously isolated group re-localized by the Image-Cytometry Analysis and acquired by Structured Illumination Microscopy (SIM). 53BP1: Magenta; DYNLL1: Red; PLA: Yellow; DNA: Cyan. Scale bar: 5 µm.

**Figure 6 cells-12-00354-f006:**
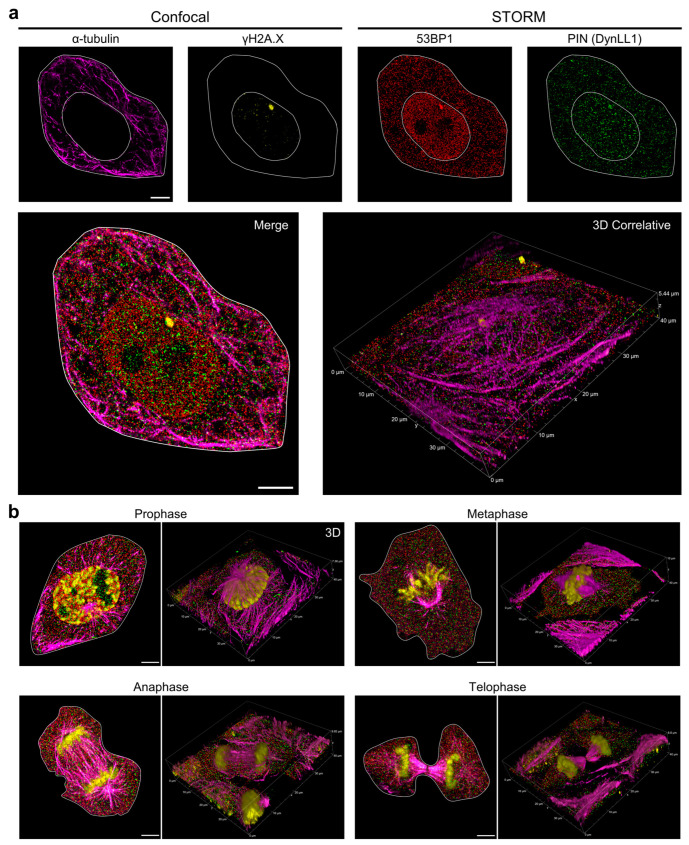
Correlative Confocal-dSTORM Analysis on Image-Cytometry re-localized cells. (**a**) Representative re-localized G1 cells acquired by the correlative Confocal-STORM protocol described in the text. Confocal Z-Stacks were first acquired of the Tubulin-Atto425 and γH2A.X-Alexa488 fluorescence channel, followed by acquisition of the 53BP1-Cy3 and DynLL1-Alexa Fluor647 channels in dSTORM. The dSTORM plane is identified in the Z-Stack, aligned and merged in the plane (merged image in the second row on the left) and then inserted into the Z-stack (3D projection on the right). (**b**) The described correlative microscopy procedure was applied to cells in the different stages of mitosis, isolated according to the image-cytometry analysis. Z-stacks merged with the dSTORM plane of some representative cells are shown. Scale bar: 5 µm.

**Figure 7 cells-12-00354-f007:**
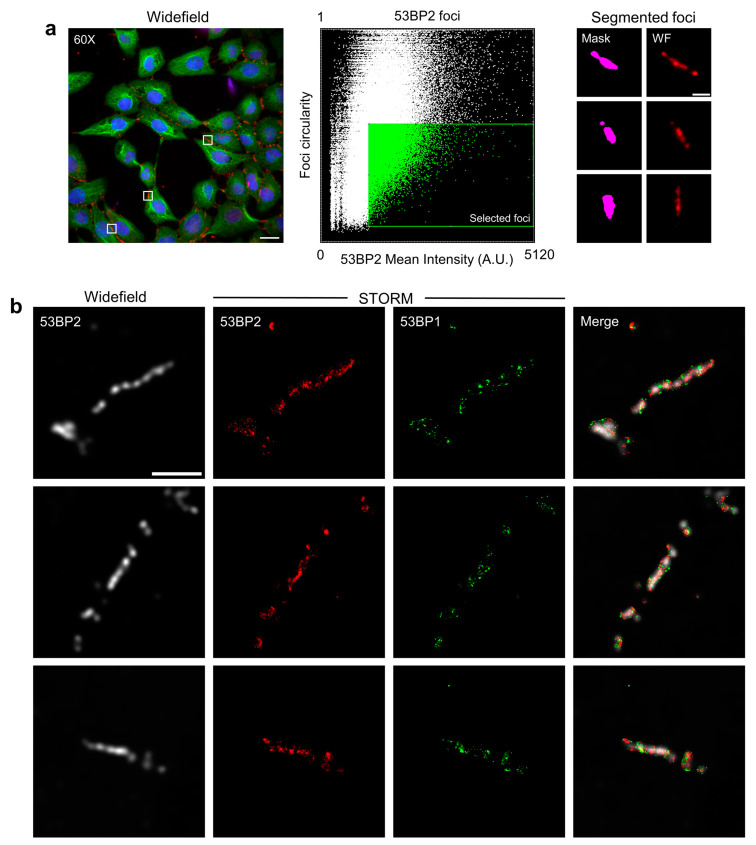
53BP2 clusters localized at the cell membrane by Image-Cytometry Analysis. (**a**) 53BP2 shows accumulation in elongated structures that can be identified by high mean intensity per pixel and circularity when considering their distribution independently from the cell of origin (dot plot in the middle column). Scale bar: 20 µm (Widefield), 3 µm (53BP2 foci). (**b**) Shown are representative images of widefield 53BP2 elongated clusters, re-localized by the intensity-shape analysis, and re-acquired by dSTORM (first column: 53BP2 widefield image, second column: 53BP2, third column: 53BP1, fourth column: merged channels). Scale bar: 3 µm.

**Figure 8 cells-12-00354-f008:**
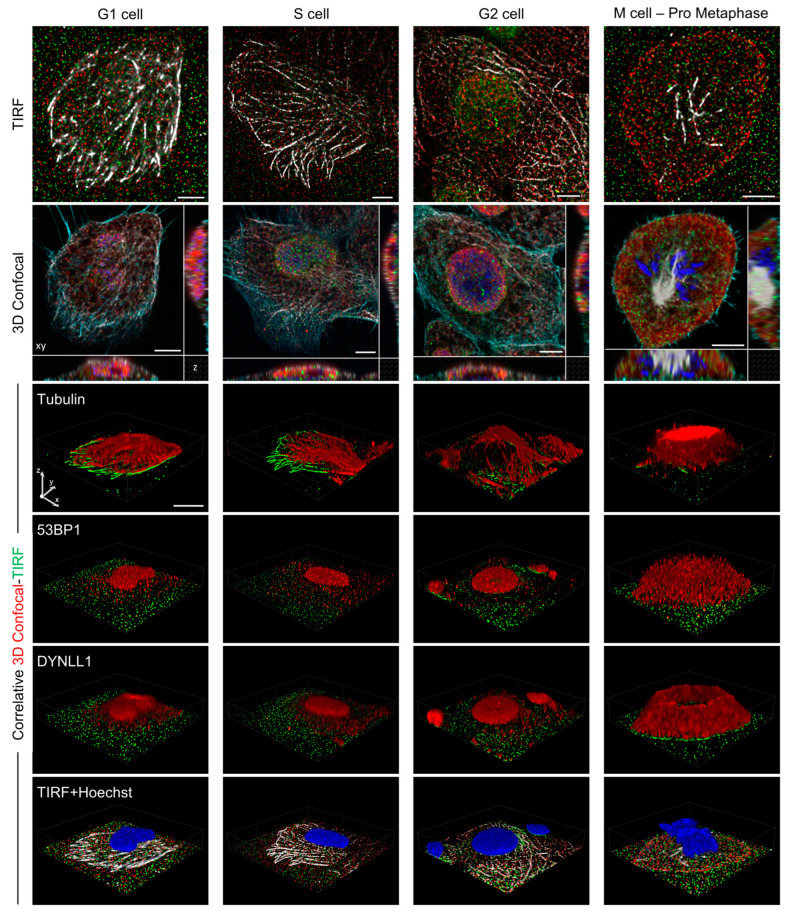
Correlative Confocal-TIRF Analysis on Image-Cytometry re-localized cells. Representative re-localized cells in different cell-cycle phases (G1, S, G2, M) acquired by the correlative Confocal-TIRF protocol described in the text. TIRF images of 53BP1, DynLL1, and Tubulin were first acquired, followed by Confocal Z-Stacks of 53BP1 (Green), DynLL1 (Red), Tubulin (White), Phalloidin (Cyan), and Hoechst (Blue). The TIRF plane is identified in the Z-Stack, aligned, and merged at the corresponding position and then inserted into the Z-stack (3D projections). The rows show (from top to bottom): TIRF images, lateral planes reconstructions of Confocal Z-Stacks (scale bar: 5 µm), and correlative 3D volume projections (scale bar: 10 µm) of Tubulin, 53BP1, DynLL1, and Hoechst. Three-dimensional projections show the proteins distribution at the basal membrane and in the cell volume (red: confocal; green: TIRF).

**Figure 9 cells-12-00354-f009:**
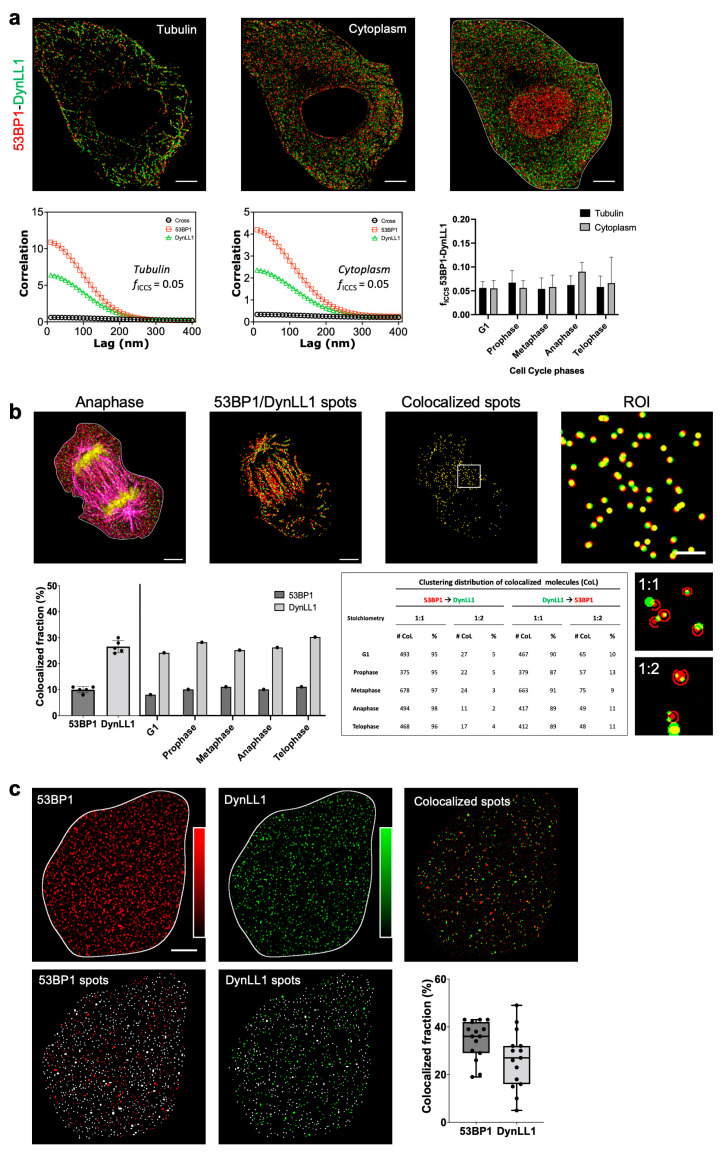
53BP1 and DynLL1 colocalization analysis of dSTORM (**a**,**b**) and TIRF (**c**) images of MCF10A cells. (**a**) (Upper row) Dual-color dSTORM image of the protein distributions (from left to right) on the cytoskeleton (tubulin), in the cytoplasm, and in the whole cell. (Lower row) Spatial correlation functions of depicted images recovered by ICCS and colocalized fraction (fICCS) extracted from ICCS analysis at different cell-cycle phases (data are mean ± s.d. of the mean values of fICCS calculated on each cell on tubulin and cytoplasm compartments; n = 25 cells). The ICCS plot shows the cross-correlation function (black squares) and the red (red circles) and green (green triangles) show channel autocorrelation functions along with the corresponding fits (solid lines). (**b**) A.M.I.CO image analysis of 53BP1 and DynLL1 spots distribution in the tubulin compartment (pink) on a representative 2D correlative dSTORM-Confocal image of a MCF10A mitotic cell. The mean colocalized fraction of 53BP1 and DynLL1 spots (data are mean ± s.d. of the mean values of the fraction of colocalized spots with respect the total amount of molecules detected in each cell; n = 5 cells) and the level of colocalization for each cell-cycle phase are shown. The table shows the clustering distribution of colocalized 53BP1 and DynLL1 molecules: data mostly reveal a 1:1 binding ratio between molecular species, calculated in the <50nm range, as depicted in representative clustering images. (**c**) A.M.I.CO image analysis of 53BP1 and DynLL1 spots distribution at the basal membrane of representative TIRF images of MCF10A cells. Green and red spots represent the colocalized fraction of the total number of molecules (grey) for each channel. The mean colocalized fraction of 53BP1 and DynLL1 spots is shown (data are mean ± s.d. of the mean values of the fraction of colocalized spots with respect to the total amount of molecules detected in each cell; n = 15 cells). Scale bar: 3 µm. Scale bar ROI: 1 µm.

## Data Availability

Raw data are available upon request.

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
