# Peer review of "Correlative Multi-Modal Microscopy: A Novel Pipeline for Optimizing Fluorescence Microscopy Resolutions in Biological Applications"

_cells, 2023, doi:10.3390/cells12030354_

Round 1

Reviewer 1 Report

The manuscript is to a large extent a detailed protocol for multi-modal microscopy with a commercial top-tier microscope. The authors rightly argue that in modern microscopy, the different methods can individually be optimized for sensitivity, spatial resolution, temporal resolution, 3D visualization while several modalities should be combined in order to both, say, have high spatial resolution and sensitivity in a 3D visualization of a cell with specific labelled constituents. In addition, good statistics (by imaging many single cells) is preferable.

The manuscript presents example studies of molecular complexes of 53BP1 protein in MCF10A cells, demonstrating an automatized process based on a previously published image-cytometry computational platform (A.M.I.C.O) and the high-end commercial microscope.

The paper is well written and well structured, figures and images are clear, and the example protocol should be interesting to a large group of readers of Cells.

Author Response

We are grateful to the reviewer for the appreciation of our work. 

Reviewer 2 Report

The paper describes a multimodal imaging pipeline, which combines confocal, TIRF, and various superresolutuon microscopy techniques. The idea is to obtain an improved characterization of cellular samples.

My problem with the manuscript is that it hardly goes beyond a descriptive presentation of the capabilities of multimodal imaging, combined with a collection of representative images. What is the advance compared to the state of the art? Further, the introduction doesn't provide much information on the state of the art of multimodal imaging, the issues of image registration, the advantages of a multimodal approach. Finally, also the biological applications did not go beyond a collection of images.   

Reviewer 3 Report

In this manuscript, Pelicci et al described a microscope system for correlative, multiscale fluorescence imaging of fixed cell samples. They developed a protocol to first identify cells of interest by widefield scanning, and then to refine the resolution combining confocal, 2D-STORM, SIM, or TIRF microscopy. As a demonstration of the method, the authors revealed the subcellular distribution of 53BP1 and DynLL1 across different cell-cycle phases to evaluate their roles of DNA damage response. The described method is of potential interest to the cell biology community because it leverages the trade-off between the cost of time for high-resolution microscopy and the throughput of cells. The data quality is fine. Writing is mostly clear with a few confusing points as listed below. However, my major critique is the lack of data analysis of colocalization for the STORM and TIRF datasets. These results were presented in a vague, qualitative way that was not more informative than the PLA confocal itself. The authors need to analyze the data more to demonstrate the usefulness of super-resolution techniques in this particular question. Things to look into include the fraction of colocalizing molecules and the size distribution of interacting domains.

Minor points:

1. The immunostaining method seemed to involve two rounds of primary/secondary cycles in addition to the PLA protocol. Why are the two rounds necessary? 

2. The authors should discuss the labeling and color crosstalk. There were dye-conjugated primary antibody labeling after a previous round of primary/secondary. Will the latter primary bind to the former secondary causing mislabeling? In some cases, the authors had 9 color channels, and I believe that there was some extent of color crosstalk between channels. These should be discussed in a more concrete way instead of saying all crosstalks were "avoided". 

3. I don't understand how the PLA worked in the manuscript. Why did the authors use dye-conjugated antibodies instead of oligo-conjugated ones? Or did they just omit all the steps for PLA?

4. What was the detector of confocal?

5. Line 262: 1.49 NA "objective".

6. Line 269: The color channels should be described with fluorophores rather than target proteins.

7. The STORM buffer does not seem to have glucose. How does the glucose oxidase work then? Also, please described the final concentration of the reagents. 

8. During scanning or switching between different modes, how did the authors manage to maintain enough amount of oil on the objective as oil can be adsorbed by the glass surface?

9. Figure 2: "basement membrane" should better be "basal membrane".

10. Figure 4b: there was no data showing the colocalization between PLA and gamma-H2AX, although it was said so in the text.

11. Line 607: what does "resolution limited" mean here? Is 120 nm higher or lower than the SIM resolution?

12. Figure 8: there was no color annotation for the top rows of images.

13. Some citation formats are inconsistent/missing and needs proofreading.

Round 2

Reviewer 2 Report

Since the authors hardly addressed my previous questions and hardly changed the manuscript upon my criticism, my recommendation remains the same. I still consider this paper as a mere collection of images, without much scientific merit. In their rebuttal letter the authors state that the aim was to "present a novel imaging pipeline focusing on the methodology". But there is nothing new on the methodology side; they just applied a commercial microscopy system. This manuscript would fit the requirements of an application note, not of a scientific paper.